# Jejunal Transcriptomic Profiling for Differences in Feed Conversion Ratio in Slow-Growing Chickens

**DOI:** 10.3390/ani11092606

**Published:** 2021-09-05

**Authors:** Panpradub Sinpru, Cindy Riou, Satoshi Kubota, Chotima Poompramun, Wittawat Molee, Amonrat Molee

**Affiliations:** School of Animal Technology and Innovation, Institute of Agricultural Technology, Suranaree University of Technology, Nakhon Ratchasima 30000, Thailand; panpradub.s@g.sut.ac.th (P.S.); cindyriou@sut.ac.th (C.R.); skubota@sut.ac.th (S.K.); cp.chotima@gmail.com (C.P.); wittawat@sut.ac.th (W.M.)

**Keywords:** feed conversion ratio, jejunum, intestine, digestive tract, transcriptome, RNA sequencing, slow-growing chicken, feed efficiency, poultry production

## Abstract

**Simple Summary:**

The slow-growing Korat chicken (KR) is economically attractive, as KR meat has a high selling price and has thus been used in Thailand to support smallholder farmers. However, low feed efficiency in KR stockbreeding makes the product less competitive and improving KR feed efficiency is central to increasing KR profitability. Using RNA sequencing, we compared the jejunal transcriptomic profiles of low- and high-feed conversion ratio (FCR) KR chickens, to identify FCR-related transcriptional variation and biological pathways. Gene Ontology and Kyoto Encyclopedia of Gene and Genome analysis revealed that the main pathways involved in KR FCR variation are related to immune response, glutathione metabolism, vitamin transport and metabolism, lipid metabolism, and neuronal and cardiac maturation, development, and growth. This is the first study to investigate, in the jejunum, the molecular genetic mechanisms affecting the FCR of slow-growing chickens. These findings will be useful in line-breeding programs to improve feed efficiency and profitability in slow-growing chicken stockbreeding.

**Abstract:**

Improving feed efficiency is an important breeding target for the poultry industry; to achieve this, it is necessary to understand the molecular basis of feed efficiency. We compared the jejunal transcriptomes of low- and high-feed conversion ratio (FCR) slow-growing Korat chickens (KRs). Using an original sample of 75 isolated 10-week-old KR males, we took jejunal samples from six individuals in two groups: those with extremely low FCR (*n* = 3; FCR = 1.93 ± 0.05) and those with extremely high FCR (*n* = 3; FCR = 3.29 ± 0.06). Jejunal transcriptome profiling via RNA sequencing revealed 56 genes that were differentially expressed (*p* < 0.01, FC > 2): 31 were upregulated, and 25 were downregulated, in the low-FCR group relative to the high-FCR group. Functional annotation revealed that these differentially expressed genes were enriched in biological processes related to immune response, glutathione metabolism, vitamin transport and metabolism, lipid metabolism, and neuronal and cardiac maturation, development, and growth, suggesting that these are important mechanisms governing jejunal feed conversion. These findings provide an important molecular basis for future breeding strategies to improve slow-growing chicken feed efficiency.

## 1. Introduction

In poultry breeding, improving feed efficiency—the efficiency of converting energy and nutrients from feed into tissue—presents an important environmental and economic challenge [1]. Low feed efficiency raises production costs and reduces competitiveness, particularly when combined with unstable feed costs [2], and improving feed efficiency could increase profitability. Feed efficiency is commonly measured in poultry production using the feed conversion rate (FCR, the ratio of feed intake to body weight gain [3,4]), especially in the production of chickens for meat [5,6]. In male slow-growing broilers, selecting for higher FCR can effectively increase feed efficiency and improve the growth rate and market weight, without affecting carcass composition [7]. However, FCR is also highly associated with production traits; some studies show that selecting for lower FCR increases the body weight gain, but it can also increase the feed intake, increase the average daily gain, and reduce the meat quality [7,8]. It is thus difficult to improve FCR via traditional breeding. Therefore, understanding the molecular basis for FCR is necessary to improve poultry production.

The slow-growing Korat chicken (KR) is produced from an indigenous breed that retains its ancestral behavioral and phenotypic traits [9]. The KR line—a crossbreed between a male of a Thai indigenous chicken line (Leung Hang Khao) and a female of a broiler line (Suranaree University of Technology)—was established to support smallholder farmers, to ensure food security in communities, and to contribute to preserving indigenous chicken breeds. KR meat has a unique taste—compared to broiler meat it is firmer and chewier, with lower fat and higher collagen content, giving it a higher selling price [10,11]. KR chickens have an average daily weight gain of 19.8–21.0 g/d, and FCR of 2.2–2.3 [12]; they are sent to the market at ca. 1.2–1.7 kg bodyweight, at 10 weeks of age.

Digestive efficiency is important in determining feed efficiency, and is highly heritable [13]. The digestive system is essential for the conversion of ingested food into the nutrients required for growth, maintenance, and reproduction [14]. The conversion of energy and nutrients from feed into tissue depends partly on nutrient absorption [15]. The jejunum is an important small intestine section that ensures nutrient absorption [16,17].

RNA sequencing (RNA-seq) has been used widely and successfully to examine economically important traits in many cultured species such as carp [18], sheep [19], cows [20], and chickens [21]. Many studies have been conducted on poultry feed efficiency using RNA-seq. However, most have focused on the selection on residual feed intake [21,22,23,24]; few data are available on FCR selection. A prior study in ducks reported that FCR selection is associated with significant gene expression variation [25]. Recent RNA-seq studies have been conducted on native chickens [23,24], dwarf chickens [21], divergent lines [26], and broiler lines [22,27]. These report that selection for a higher feed efficiency induces gene expression variation in the chicken muscle [22,24,27], duodenum [21,22,23], and jejunum [22], for which few data are available. Juanchich et al. [26] examined gene expression in the gizzard and jejunum of broiler chickens in a line divergently selected for their digestive efficiency. They did not find differentially expressed genes (DEGs).

To the best of our knowledge, no data have yet been published on the effect of FCR selection on jejunal gene expression in slow-growing chicken. The objective of this study is to examine the effect of FCR selection on the jejunal gene expression in slow-growing chickens. Our ultimate goal is to find candidate genes that can improve FCR. Here, using RNA-seq to compare the jejunal transcriptome profiles of low- and high-FCR KR groups, we examined the biological pathways connecting FCR and feed efficiency. We identified 56 genes related to the FCR that are enriched in biological processes related to immune response; the metabolism of glutathione, vitamins, and lipids; and neuronal and cardiac maturation, development, and growth. These genes are potentially central to governing jejunal feed conversion. These findings will improve understanding of the mechanisms underlying feed efficiency in slow-growing chickens and will improve breeding selection programs aimed at producing slow-growing chickens while minimizing feed costs.

## 2. Materials and Methods

### 2.1. Ethics Statement

The animal handling and maintenance procedures used in the present study were approved by the Ethics Committee on Animal Use of Suranaree University of Technology, Nakhon Ratchasima, Thailand (permit number: U1-02631-2559).

### 2.2. Experimental Animal and Tissue Collection

We used 75 one-day-old male slow-growing KR chickens. They were raised in individual cages under a 16L:8D light regimen and fed ad libitum three types of commercial feed (CPF Co., Ltd., Nakhon Ratchasima, Thailand): a starter diet (21% protein), a grower diet (19% protein), and a finisher diet (17% protein) at 0–3, 4–6, and 7–10 weeks of age, respectively. An automatic nipple watering system was installed individually in each cage, and water was freely available. The FCR was calculated as previously described [7], using the following equation:(1)$$\mathrm{FCR}=\frac{FI}{BWG}$$
where *FI* represents the total feed intake from 1 week to 10 weeks (g), and *BWG* is the body weight at 10 weeks of age (g) minus the body weight at 1 week (g).

The chickens were then ranked by FCR at 10 weeks of age. Six individuals with extreme FCR values were chosen: three with extremely low FCR (1.83–1.99), and three with extremely high FCR (3.18−3.36). After 8 h of fasting, these six males were stunned using chloroform for knockout and were sacrificed by neck cutting for bleeding. The jejunum was dissected immediately from the carcass, cut into 1 cm segments, snap-frozen in liquid nitrogen, and subsequently stored at −80 °C until further processing. The significance of differences in growth performance between the FCR groups was determined using Student’s *t*-test. The statistical significance threshold was set at *p* < 0.05.

### 2.3. RNA Extraction

Total RNA was extracted from each sample using TRIzol reagent (Thermo Fisher Scientific, Carlsbad, CA, USA). Jejunal tissue from each bird was lysed and homogenized in TRIzol reagent. After centrifugation, the supernatants were transferred to new tubes and incubated with chloroform for 5 min. Samples were then centrifuged for 10 min at 12,000× *g* (Thermo Fisher Scientific, Langenselbold, Germany). Pellets were precipitated using isopropanol, washed with 75% ethanol, and dried at 25 °C for 5 min. RNA pellets were resuspended in 20 µL of nuclease-free water. The quantity and quality of the extracted RNAs were monitored using an Agilent 2100 Bioanalyzer (Agilent Technologies, Palo Alto, CA, USA), a NanoDrop (Thermo Fisher Scientific, CA, USA) device, and a 1% agarose gel. One microgram of total RNA with RIN > 7.5 was used for library preparation.

### 2.4. RNA Library Preparation and Sequencing

RNA-seq library preparation, sequencing, and analysis were carried out by Vishuo Biomedical Ltd. (Bangkok, Thailand), following the procedure of [28]. For RNA-seq preparation, poly (A) mRNA isolation was performed using the NEBNext^®^ Poly(A) mRNA Magnetic Isolation Module (New England BioLabs, Ipswich, MA, USA). mRNA fragmentation and priming were performed using NEBNext^®^ First Strand Synthesis Reaction Buffer and Random Primers (New England BioLabs, Ipswich, MA, USA). First-strand cDNA was synthesized using ProtoScript^®^ II Reverse Transcriptase (New England BioLabs). Second-strand cDNA was synthesized using the Second Strand Synthesis Enzyme Mix (New England BioLabs). Bead-purified double-stranded cDNA was treated with NEBNext End Prep Enzyme Mix (New England BioLabs) to repair both ends and to add a dA-tail in one reaction, followed by T–A ligation to add adaptors to both ends. The size selection of adaptor-ligated DNA was then performed using beads, and fragments of ~420 bp (insert size of ~300 bp) were recovered. Libraries were amplified by polymerase chain reaction (PCR) for 13 cycles. The PCR products were purified using beads, and their quality and quantity were checked using a Qsep100 (BIOptic Inc., Taiwan, China), and a Qubit 3.0 Fluorometer (Invitrogen, Carlsbad, CA, USA). The libraries were sequenced using paired-end configurations with a read length of 2 × 150 bp on an Illumina HiSeq X instrument (Illumina, San Diego, CA, USA). Image analysis and base calling were conducted using the HiSeq Control Software (HCS) + OLB + G A Pipeline-1.6 (Illumina).

### 2.5. Gene Expression and Differential Expression Analysis

To ensure high-quality data, low-quality reads and those containing adapter contamination were removed using Cutadapt v. 1.9.1 [29]. Read bases with a phred quality score less than 20 (Q20), sequencing adapters, and reads containing poly-N were filtered out to generate clean data. The guanine cytosine (GC) content of the clean reads was then calculated. All downstream analyses were based on high-quality, clean data. The reads were mapped to the chicken genome (GRCg6a, GenBank: GCA_000002315.5) using HISAT2 v. 2.0.1 [30]. Cufflinks v. 2.2.1 [31] was used to assemble the mapped reads. Alternative splicing events were extracted, quantified, and compared using ASprofile v. 1.0 [32]. Gene-level read counts were enumerated using HTSeq v. 0.6.1 [33]. Differential gene expression between high- and low-FCR samples was analyzed using the DESeq2 R package [34]. *p*-values were adjusted using a Benjamini–Hochberg correction [35] to control the false discovery rate. The criteria for identifying DEGs were FC > 2 (or |log_2_ FC| > 1), *p* < 0.01, and a *q*-value < 0.27.

The visualization of the differences and similarities between high- and low-FCR samples was performed using a principal component analysis (PCA) of the transcripts and DEGs, using the ropls R package [36]. A volcano plot (created using EnhancedVolcano in R [37]) was used to visualize the jejunal transcriptome and DEGs. A heatmap (generated using pheatmap in R [38]) was used to illustrate the DEG expression profile for each sample.

### 2.6. RNA-Seq Validation via RT-qPCR

To confirm the differential expression results, eight genes (i.e., *LY6E*, *PLAC8*, *LOC771880*, *MLKL*, *ADV*, *IFI6*, *PLA2G4B*, and *LBFABP*) were randomly selected and their expression was measured by quantitative reverse transcription PCR (RT-qPCR). Total RNA was converted into first-strand cDNA using a High Capacity cDNA Reverse Transcription Kit (Thermo Fisher Scientific, Carlsbad, CA, USA). The primers (Appendix A) for each gene were designed using NCBI Primer BLAST (https://www.ncbi.nlm.nih.gov/tools/primer-blast/, accessed on 28 October 2021) and the Ensembl genome browser (https://ensembl.org/, accessed on 28 October 2021). The amplification efficiencies for each primer pair were calculated prior to RT-qPCR validation, and the efficiency of reaction values from 90 to 110% were used for qPCR reactions. RT-qPCR reactions were conducted in triplicate on a LightCycler^®^ 480 Real-Time PCR System (Roche Diagnostics GmbH, Mannheim, Germany) using SYBR^®^ Green Chemistry. The thermocycling program consisted of an initial denaturation step at 95 °C for 10 s, followed by 45 cycles of 95 °C for 30 s, 60 °C for 30 s, and 72 °C for 1 min, with a final extension at 72 °C for 5 min. Relative gene expression was calculated using the 2^−Δ^^ΔCt^ method, using the housekeeping gene *GAPDH* as an internal control [31]. To validate the RNA-seq results, we conducted linear regression analysis of the log_2_ FC scores of the RNA-seq and RT-qPCR analyses using a regression analysis tool from the data analysis tool pack of Microsoft^®^ Excel^®^ 2016 (Microsoft Corp., Redmond, WA, USA). The correlation between log_2_ FC was calculated with the Pearson test using SPSS v. 24.0 for Windows software (SPSS Inc., Chicago, IL, USA). The statistical significance threshold was set at *p* ≤ 0.01.

### 2.7. GO Term and KEGG Pathway Enrichment Analysis

Functional annotation and Gene Ontology (GO) term enrichment were performed using the Ensembl and Entrez Gene (NCBI) databases for *Gallus gallus*, using ViSEAGO [39] in R. The dataset of the jejunum-expressed genes was used as a background for DEG GO term enrichment (Fisher’s exact test, *p*-value < 0.05). Multi-dimensional scaling (MDS) plots constructed from the semantic similarity distances between enriched GO terms were generated using ViSEAGO [39].

Pathway enrichment analysis was based on Kyoto Encyclopedia of Genes and Genomes (KEGG) pathway units. A hypergeometric test was used to identify the DEG pathways that were significantly enriched against the transcriptome background, using the following formula:(2)$$p=1-\overset{m-1}{\underset{i=0}{\sum }}\frac{\left(\begin{array}{c} M\\ i \end{array}\right)\left(\begin{array}{c} N-M\\ n-i \end{array}\right)}{\left(\begin{array}{c} N\\ n \end{array}\right)}$$
where *N* is the number of genes with pathway annotations; *n* is the number of DEGs in *N*; *M* is the number of genes annotated for a particular pathway in all genes; and *m* is the number of DEGs annotated for this pathway.

A scatter plot was used to graphically represent the KEGG enrichment analysis results. The degree of enrichment was assessed using the Rich factor (the ratio of the number of DEGs in the pathway to the total number of genes in the pathway), *q*-value < 0.05, and the number of genes enriched in each pathway.

## 3. Results

### 3.1. Feed Efficiency Associated with Low- and High-FCR

At 10 weeks of age, FCR was significantly lower in the low FCR group (1.93 ± 0.05) than in the high-FCR group (3.29 ± 0.06) (*n* = 3 per group; *p* < 0.05; Table 1). The low-FCR group had significantly lower feed intake and significantly higher body weight gain than the high-FCR group (*p* < 0.05).

### 3.2. Genome Mapping Statistic

RNA-seq of the six jejunum epithelial samples generated 264.9 million raw reads (Table 2). After filtration, 6.3 Gb on average of high-quality data (Q20 > 95%, Q30 > 90%) was retained for each sample. The filtered data were subsequently aligned with the reference genome. Over 83% of the clean reads per sample were mapped to the *Gallus gallus* genome assembly. Between 78.15% and 80.49% reads were uniquely mapped, whereas 4.76–7.84% reads were mapped more than once.

Mapped reads were assigned to genomic features, exons, introns, and intergenic regions. Most of the sequences (59.09–70.04%) were mapped to exonic regions; 13.34–21.38% to intergenic regions; and 12.12–27.57% to intronic regions (Figure 1). This indicates that our sequences matched the reference genome mainly in coding regions and were therefore acceptable for further analysis.

### 3.3. Differential Expression Profiling

In total, 24,356 transcripts were identified in the jejunum samples. Based on the PCA (Figure 2), high-FCR samples were closely grouped together, whereas low-FCR samples were scattered, reflecting natural biological variation in gene expression in the low-FCR group. Differential gene expression analysis revealed 56 DEGs (*p* < 0.01, log_2_ FC > 1), of which 31 were upregulated and 25 downregulated in the low-FCR group relative to the high-FCR group (Figure 3). Those genes differentially expressed in response to differences in FCR are summarized in Table 3. Among the low-FCR individuals, individual L2 had a distinct pattern of DEG expression (Figure 4).

We conducted PCA of the DEGs to examine the high level of natural variation in the low-FCR group. There was a distinct separation between the high- and low-FCR groups (Figure 5), providing evidence that these DEGs are appropriate for separating these groups, in spite of the natural variation in the low-FCR group.

To validate these results, eight DEGs (i.e., *LY6E*, *PLAC8*, *LOC771880*, *MLKL*, *ADV*, *IFI6*, *PLA2G4B*, and *LBFABP*) were randomly selected for RT-qPCR assays using the same RNA samples used for RNA-seq. The linear regression of RNA-seq and RT-qPCR log_2_ FC scores (Figure 6) reveals that these selected DEGs showed concordant expression patterns using both methods, with a strong positive association (R^2^ = 0.9875). Moreover, Pearson’s correlation also showed that this association is significantly correlated (*r* = 0.960, *p* < 0.01). Thus, the RT-qPCR analysis validated the RNA-seq results.

### 3.4. Functional Annotation and GO Term Enrichment Analysis

The significantly enriched Ensembl-derived GO terms are shown in Figure 7. Appendix A shows the associated DEGs, and Table 4 shows the significantly enriched Entrez Gene-derived GO terms. GO term enrichment revealed 22 GO terms in the BP category, 6 in the MF category, and none in the CC category. The Ensembl-derived MDS plots, which arrange the GO terms into several main groups, reveal the most important functions associated with jejunal differences in FCR (Figure 7A). Based on the Ensembl-derived MDS plots and the Entrez Gene-derived GO terms, the main biological processes implicated are: immune response; neuronal and cardiac maturation, development and growth; and glutathione and vitamin metabolism.

### 3.5. Pathway Enrichment Analysis

The top 30 most significantly enriched KEGG pathways are shown in Figure 8, and Appendix A shows the associated DEGs. The pathways for vitamin digestion and absorption, and the primary immunodeficiency pathways, had the highest Rich factors, indicating their importance in jejunal differences in FCR.

Consistent with the GO term analysis, KEGG pathway enrichment analysis revealed that the main biological pathways involved in jejunal differences in FCR were those related to immune response; neuronal and cardiac maturation, development and growth; glutathione and vitamin metabolism; and lipid metabolism.

## 4. Discussion

The jejunum is the primary site of nutrient absorption [16,17]. The jejunum transcriptome has been studied previously in broiler lines with differences in digestive efficiency [26] and ducks with differences in FCR [25], an indicator of feed conversion efficiency. Our study represents the first analysis of jejunal transcriptomic differences associated with FCR in slow-growing chickens. We investigated DEG–associated functional networks to elucidate feed conversion and feed efficiency in slow-growing chickens.

For the KR chickens that we studied, FCR was 1.93 in the low-FCR group and 3.29 in the high-FCR group. In a KR population, the market-age FCR averages 2.2 [12], which is below that of commercial slow-growing meat-type chickens, at 3.22 [40], and Chinese yellow slow-growing chickens, at 3.15 [7,40]. We observed that, for KR chickens, higher body weight can be achieved with reduced feed intake, even at FCR values as low as 1.93. In Chinese slow-growing chickens, selecting for FCR can improve the market weight, even though FCR is negatively correlated with feed intake [7,21]. This suggests that the growth dynamics of KR chickens may be unique among slow-growing chickens.

Our functional annotation of DEGs revealed an enrichment of biological processes related mainly to immune response, glutathione metabolism, vitamin transport and metabolism, lipid metabolism, and neuronal and cardiac maturation, development, and growth. This suggests that these might be important mechanisms governing jejunal feed conversion in KR chickens.

### 4.1. Immune Response

Immune defense is central to intestinal function, because the intestinal epithelium is in contact with the feed and microbiota [41]. Although we did not select for immune-related traits, immune response was the most important enriched pathway, representing various DEGs. Consistent with many research studies on poultry, in which selecting for higher feed efficiency was found to be associated with muscular and intestinal epithelial immune response [22,24,25,27], we found that FCR in KR chickens is associated with differences in jejunal immune response.

Within the innate immune system, macrophages are essential for maintaining mucosal homeostasis and epithelial renewal [42]. Jejunal tissue is enriched in genes related to immune defense and immune response via macrophages [26], highlighting the importance of macrophages in the immune system of the chicken jejunum. Two DEGs, *SPON2* (Spondin-2) and *MMP10* (Matrix metalloproteinase 10, or stromelysin 2), with higher expression in the low-FCR group, are involved in macrophage activation and function. In contrast, in the muscle of high feed efficiency chicken, a decrease in macrophage activation has been predicted [22]. *SPON2* encodes a protein that binds directly to bacterial lipopolysaccharide (LPS), functioning as an opsonin for macrophagic phagocytosis of bacteria [43]. In the jejunum of steers, genes related to bacterial LPS were more highly expressed in high feed efficiency (high gain–low intake) individuals than in low feed efficiency (low gain–high intake) individuals, suggesting that high feed efficiency is associated with higher macrophage bactericidal potential [44]. Therefore, higher *SPON2* expression in the low-FCR group may indicate a greater ability to eliminate bacterial LPS via the macrophage pathway.

*MMP10* encodes a critical cell-autonomous mediator controlling macrophage activation [45]. Our results suggest that higher *SPON2* and *MMP10* expression might ensure the maintenance of a healthy environment in the low-FCR jejunum, with a greater potential to eliminate bacteria and other undesirable particles. Their lower expression in the high-FCR group highlights the importance of this pathway for feed efficiency. Therefore, higher *SPON2* and *MMP10* expression might be essential for improving feed efficiency.

Several of the DEGs are related to T-cell activation (i.e*., LAPTM4B*, *LOC771880*, *LAG3*, and *LY6E*). T-cells are essential in the adaptive immune response, producing interferons and interleukins to coordinate an appropriate immune response [46]. Intestinal T-cells are critical in control and protection against pathogenic infections [47]. T-cell proliferation is associated with improved growth performance in animals treated with feed supplements and provides a protective immune response [48,49]. *LAPTM4B*, which encodes a protein that negatively regulates active transforming growth factor beta 1 (TGF-β1) production in human regulatory T-cells (Tregs) [50], was expressed more highly in the low-FCR group. The production of active TGF-β1 is one mechanism whereby human Tregs suppress immune responses [51]. Thus, higher *LAPTM4B* expression in the low-FCR jejunum may lead to T-cell activation, which would not occur to the same extent in the high-FCR group.

*LOC771880* (*CD8A*) and *LAG3* were more highly expressed in the high-FCR group. *CD8* encodes a cell-surface glycoprotein (antigen) found on most T-cells [52,53]. *LAG3* encodes a protein that functions as a T-cell inhibitor [54,55,56]*. CD8*+ T-cells are precursors of cytotoxic T-cells, which are efficient immune effectors with an important role in eliminating virus-infected cells [57]. *CD8*+ T-cell proliferation is associated with higher growth performance in pigs with viral infection [48]. The *CD8A* expression profile in the high-FCR jejunum may thus indicate that its immune system is potentially more sensitive to viral infection. This is supported by our finding that *LY6E*, which promotes viral entry in human cell lines [58], was more highly expressed in the high-FCR group. Together, our findings reveal that the lower T-cell inhibitor expression in the low-FCR jejunum may promote T-cell activation, which is essential in the adaptive immune response. In contrast, in the muscle of native chickens selected on residual feed intake, genes relative to T-cell activation were downregulated in high feed efficiency chickens [24].

Changes in the cell death process may lead to severe disorders, including inflammatory diseases; this crucial process is therefore finely controlled in the intestinal barrier [59]. Previous studies have reported that selection for higher feed efficiency in chickens involves variation in the expression of muscular genes related to cell death [24,27]. Apoptosis and necroptosis are programmed forms of cell death involved in intestinal barrier homeostasis and renewal [59]. Two of the DEGs were related to apoptosis and necroptosis pathways. *PLAC8*, which was more highly expressed in the high-FCR group, was first identified via a microarray analysis of the murine placenta [60]. Its function in apoptosis has also been demonstrated [61]. Huang et al. [62] reported that butyrate, a short-chain fatty acid produced by gut microbes, reduces *PLAC8* expression and induces apoptosis in the human colon. A previous study in commercial broiler chicken showed that muscular genes related to cell death and survival were upregulated in a high feed efficiency group [27]. In contrast, a recent study in native chickens found an upregulation of muscular genes related to the apoptosis pathway in a low feed efficiency group [24]. We propose here that the higher expression of *PLAC8* in the high-FCR jejunum may reduce apoptosis, thus compromising jejunal barrier homeostasis and renewal.

In contrast to the *PLAC8* expression profile, *MLKL* was more highly expressed in the low-FCR group. *MLKL* mediates TNF-induced necroptosis [63,64,65,66], and the *RIPK3*–*MLKL* pathway is suggested to be important for activating necroptosis in digestive organs [67]. The higher *MLKL* expression in the low-FCR jejunum may promote necroptosis, ensuring jejunum barrier homeostasis and renewal, in contrast to the effect of *PLAC8* in the high-FCR jejunum. *PLAC8* and *MLKL* expression thus appears to be important for regulating homeostasis and renewal of the KR jejunal epithelium, ensuring epithelial integrity, and thereby ensuring a suitable environment for feed absorption.

### 4.2. Glutathione Metabolism

Higher oxidative stress is associated with low feed efficiency in broilers [68], steers [44], and cattle [69]. The antioxidant system maintains a diverse microbiota in the luminal epithelia of the gastrointestinal tract [70]. Production of reactive oxygen species and reactive nitrogen species by gastrointestinal epithelial cells or enteric commensal bacteria causes intestinal inflammation and impairs absorption [70]. Our findings revealed two DEGs (*MMACHC* and *CHAC1*) associated with glutathione metabolism; they are closely involved, respectively, in vitamin metabolism and neuronal function. *MMACHC*, more highly expressed in the low-FCR group, encodes a protein that functions closely with vitamin B12 and its derivatives, using glutathione to generate cob(I)alamin (a vitamin B12 derivative) and glutathione thioether [71]. Given that the formation of a thioether derivative in the glutathione pathway serves to detoxify xenobiotic compounds [72], our findings suggest that the low-FCR jejunum may have greater cellular detoxification potential than the high-FCR jejunum. *MMACHC* functions as a glutathione S-transferase. A previous study in a native chicken showed an upregulation of several glutathione S-transferase in the muscle of high feed efficiency native chickens, suggesting that responses to oxidative stress in high feed efficiency chickens is elevated [27]. In agreement with this, the findings of two previous studies suggest that chickens with high residual feed intake (low feed efficiency) are more susceptible to oxidative stress, since an upregulation of several muscular and duodenal genes responsible for ROS production in low efficiency animals was found [21,24].

*CHAC1*, also more highly expressed in the low-FCR group, encodes a protein that catalyzes glutathione cleavage into 5-oxo-l-proline and a Cys-Gly dipeptide, functioning as a glutathione-degrading enzyme [73,74]. Higher *CHAC1* expression may thus sensitize low-FCR jejunal cells to oxidative injury; nevertheless, other pathways may balance redox homeostasis. Moreover, glutathione degradation by *CHAC1* may represent an important novel pathway in neuronal development and pathogenesis [75].

### 4.3. Vitamin Transport and Metabolism

Vitamin binding and metabolism pathways were enriched in the jejunal DEGs. In the beef steer jejunum, vitamin binding-related genes were significantly enriched in a high-feed efficiency group [44], indicating that this biological pathway related to vitamin binding may be important for feed efficiency. *AVD*, which was more highly expressed in the high-FCR group, encodes a protein that binds to biotin (water-soluble vitamin B8). *AVD* is localized in chicken intestinal goblet cells; given that its expression increases in response to bacterial LPS stimulation, it may serve as an antibacterial mucus layer in the intestinal epithelium [76]. Variation in the composition of the jejunal mucus layer may be highly important, considering that mucus layer properties are important for the absorptive function of the small intestine [77]. However, considering that the gene expression profiles of the high-FCR individuals might reduce their immune response, the higher *AVD* expression may ensure a minimal defense system to protect the epithelium. *AVD* and *SPON2* (higher expressed in the low-FCR group) are related to the bacterial LPS degradation pathway [43,76]. Therefore, considering their contrasting expression profiles, we propose that the differences in FCR in KR chickens may be associated with different pathways responding to bacterial LPS stimulation.

*MMACHC*, related to glutathione metabolism and more highly expressed in the low-FCR group, encodes a cytosolic chaperone responsible for the processing and intracellular trafficking of cobalamin (water-soluble vitamin B12) by participating in the conversion of vitamin B12 into adenosylcobalamin (AdoCbl) and methylcobalamin (MeCbl) [71]. AdoCbl is a cofactor of mitochondrial methylmalonyl CoA mutase, which breaks down certain protein building blocks (amino acids), fats (lipids), and cholesterol [78]. MeCbl is a cofactor of cytosolic methionine synthase, which converts homocysteine into methionine, which is used to produce proteins and other important compounds [78]. The higher *MMACHC* expression in the low-FCR jejunum may reflect a higher potential for vitamin B12 metabolism into AdoCbl and MeCbl, essential components for protein breakdown and protein synthesis, respectively. This potentially ensures jejunal epithelial renewal and the absorption and metabolism of protein from feed.

Vitamin B12 interacts with superoxide at rates similar to those of superoxide dismutase; therefore, it may protect against chronic inflammation and modulate redox homeostasis [79]. The higher *MMACHC* expression in the low-FCR jejunum may thus represent a greater potential to modulate redox homeostasis by eliminating reactive oxygen species, thereby protecting the epithelium from inflammation. *MMACHC* is therefore an important candidate for improving the feed efficiency of slow-growing KRs.

*ADH1L* (*LOC10087280*), highly expressed in the low-FCR group, participates in the metabolism of retinol (fat-soluble vitamin A) into retinoic acid, an important component that regulates gene transcription at specific DNA sites known as retinoic acid response elements (RAREs) [80]. In the murine liver, *ADH1* may minimize toxicity by rapidly metabolizing retinol into retinoic acid, thus reducing retinol utilization by P450s [81]. Retinol may induce oxidative stress and modulate antioxidant enzyme activity [82], and it plays a critical role in enhancing immune function [83]. Therefore, higher *ADH1L* expression in the low-FCR jejunum indicates that retinol metabolism may be central to improving slow-growing KR feed efficiency.

### 4.4. Lipid Metabolism

Our functional analysis revealed the enrichment of pathways related to lipid metabolism. Processes related to lipids and fatty acid B-oxidation are highly enriched in the chicken jejunum [26], indicating the importance of this pathway with regard specifically to jejunal absorption. Many studies have shown that intestinal fatty acid degradation- and synthesis-, and fat transport-, metabolism-, and absorption-related genes are associated with feed efficiency in poultry [22,23,25]. *PLAC8,* a negative regulator of apoptosis and more highly expressed in the high-FCR group, also encodes a critical upstream protein that regulates brown fat differentiation and body weight, and controls thermoregulation [84]. Brown adipocytes oxidize fatty acids to produce heat in response to cold or excessive energy intake [84]. In mice, genetic inactivation of *PLAC8* is associated with cold intolerance, late-onset obesity, abnormal morphology, and impaired brown adipocyte function [84]. Higher *PLAC8* expression in the high-FCR jejunum may therefore indicate a higher potential for brown fat differentiation to promote body weight control and thermoregulation. This may reflect higher energy expenditure in the high-FCR than the low-FCR jejunum. The lower *PLAC8* expression in the low-FCR group suggests that those individuals may be less able to adapt to changes in environmental temperature or feed intake, because of their reduced capacity for thermoregulation or body weight control.

*LBFABP*, which was more highly expressed in the low-FCR group, is predominantly expressed in the chicken’s digestive tract [85], and encodes a protein belonging to the fatty acid binding protein family. Fatty acid binding proteins are abundant cytosolic lipid-binding proteins expressed in a tissue-specific pattern [86,87]. Their expression may facilitate intracellular fatty acid trafficking from uptake to storage or oxidation, or from lipid droplets for secretion [86,87,88]. In accordance with our results, a previous research study in duck showed that several fatty acid binding proteins were upregulated in the jejunum of a low-FCR group [25], revealing the importance of the peroxisome proliferator-activated receptor (PPAR) pathway in poultry feed efficiency [23,25]. The higher *LBFABP* expression in the low-FCR group is also consistent with the findings of Prakash et al. [89], who demonstrated that *LBFABP* is upregulated in high-feed efficiency broiler duodenums. It has been suggested that *LBFABP* plays a specific role in the liver in response to food intake, since its expression is higher in high-growth than in low-growth chickens [85]. The higher *LBFABP* expression in the low-FCR group suggests greater intracellular trafficking of lipids and consequently an increased capacity for lipid utilization (including storage and oxidation). We therefore suggest that *LBFABP* may be essential in controlling feed efficiency in slow-growing KR chickens.

*PLA2G4B*, which was more highly expressed in the low-FCR jejunum encodes phospholipase A2 (in the cytosolic phospholipase A2 protein family), which hydrolyzes the sn-2 bond of phospholipids and releases lysophospholipids and fatty acids. Consistent with our results, a prior study in chickens (selected for residual feed intake) showed that *PLA2G4A* may be associated with feed efficiency, since it was upregulated in high feed efficiency duodenum [21]. Phospholipase A2 enzymes participate in membrane homeostasis by altering phospholipid composition. They also participate in energy production by supplying fatty acids for β-oxidation, in barrier-lipid generation, and in balancing saturated and unsaturated fatty acids [90]. They may also play an important role in metabolic disorders such as obesity, diabetes, hyperlipidemia, and fatty liver disease [91].

*PLA2G4B* varies in its tissue expression pattern, regulatory mechanisms, and functions. Its role in regulating metabolism remains to be clarified [91]. However, it has been implicated in age-related changes in phospholipids and in reduced energy metabolism in monocytes [92]. Moreover, the phospholipase A2 family is responsible for arachidonic acid liberation from cellular membranes. Subsequent arachidonic acid metabolism leads to the production of prostaglandins and leukotrienes—key mediators of the gut inflammatory response [90]. *PLA2G4B* might be important in improving KR feed efficiency and warrants further study.

### 4.5. Maturation, Development and Growth

Several of the identified pathways and biological processes in our study were related to neuronal and cardiac maturation, development and growth. This finding was also reported by Xiao et al. [23] and Zhou et al. [27] in chicken duodenum and muscle, respectively. In particular, *C1QL1*, *CDK5R1*, *MYOC*, and *CHAC1*, associated with neuronal function and development [93,94,95], were more highly expressed in the low-FCR group. The digestive system possesses a local nervous system (the enteric nervous system, ENS) [96,97,98], which regulates major enteric processes such as immune response, nutrient detection, microvascular circulation, intestinal barrier function, and the epithelial secretion of fluids, ions, and bioactive peptides [99]. It is therefore not surprising that jejunal genes involved in neuronal development were differentially expressed in our study in response to differences in FCR. To the best of our knowledge, the role of the ENS in feeding efficiency has not previously been examined.

*C1QL1* encodes a secreted protein proposed to regulate the number of excitatory synapses formed on hippocampal neurons [93]. *CDK5R1* encodes a neuron-specific activator of *CDK5* (cyclin-dependent kinase 5); *CDK5R1*/*CDK5* has been suggested to play a critical role in neurite outgrowth and cortical lamination [94]. *MYOC* encodes a secreted glycoprotein that regulates the activation of various signaling pathways in adjacent cells, thereby controlling numerous processes such as cell adhesion, cell–matrix adhesion, cytoskeleton organization, and cell migration. Among its many roles, *MYOC* mediates myelination in the peripheral nervous system via ERBB2/ERBB3 signaling and participates in neurite outgrowth [95]. Interestingly, recent observations of Xiao et al. [23] suggest that the upregulation of several myosins (e.g., *MYO1D*, *MYO1E*, *MYO1A*) in duodenum from high feed efficiency chickens may be related to intestine digestion and absorption function. Moreover, Zhou et al. [27] reported an upregulation of the growth-related genes myogenin (*MYOG*) and myoferlin (*MYOF*) in high feed efficiency chicken muscle. Higher *C1QL1*, *CK5K5R1*, and *MYOC* expression in the low-FCR jejunum may function in ENS neuronal development, supporting the specific absorptive function of the jejunum and thus improving feed efficiency.

*CHAC1* is responsible for the cleavage of glutathione into 5-oxo-l-proline, also called l-pyroglutamic acid (PGA), an endogenous molecule formed by glutamate cyclization [72]. PGA has been studied in metabolic disease with glutathione synthetase deficiency [100] and in neurodegenerative disease [101]. Various neurotoxic actions have been attributed to PGA [100,101,102,103]. PGA binds to glutamate receptors [104] and inhibits glutamate uptake by synaptosomes [105]. Chronic glutamate-uptake inhibition can lead to slow neurotoxicity [106]. Considering these reported negative effects, we are concerned about the endogenous use of PGA by 5-oxoprolinase. This enzyme, which catalyzes the conversion of PGA into l-glutamate, has not yet been referenced in GRCg6a (GenBank: GCA_000002315.5); we were therefore unable to identify it in the KR jejunum. However, given that it has been detected in the digestive tract of other species including humans [107], we hypothesize that it occurs in the KR jejunum, and may participate in transforming the abundant PGA in glutamate. In pigs, glutamate is central in supporting maximum growth, development, and production performance [108]. Differences in *CHAC1* expression in KR chickens might therefore affect ENS functions via the glutamate pathway, thereby crucially affecting feed efficiency.

*CHAC1* negatively regulates the *Notch* signaling pathway in embryonic neurogenesis [109]. We found that *MESP1*, which was more highly expressed in the low-FCR jejunum, was related to aspects of embryonic development such as cardiac conduction system development. *MESP1* participates in the embryonic development of the murine heart, somites, and gut [110]. In muscle from pigs with differences in residual feed intake, functional analysis of DEGs comparing high and low feed efficiency groups revealed the enrichment of various biological processes related to growth, including cardiovascular system development and function [1]. In summary, this highlights the importance of genes related to embryonic development, such as *MESP1* or *CHAC1*, in improving feed efficiency. In low-FCR KR chickens, high *MESP1* and *CHAC1* expression might be critical in establishing jejunum structure and function, both during and after embryonic development. The precise roles of *MESP1* and *CHAC1* in KR jejunum function and development remain to be elucidated.

Several limitations of this study need to be acknowledged. Despite no evidence of area-specific gene expression in the jejunum, RNA has been isolated from the entire section of the jejunum. Therefore, it is not possible to determine which area of the jejunum is associated with significant gene expression variation. The major limitation of this study is its small sample size, which made it unsuitable for statistically powerful analysis. Moreover, our study reveals biological variation in one experimental group that can also correspond to a possible batch effect, although samples have been analyzed in the same conditions with quality control steps. A surrogate variable analysis may be a useful method to remove hidden variations should any exist. Thus, these aspects limit the power of the experiment to detect DEGs. Moreover, searching for differences in DEGs between animals that share the same genetic background can represent a limitation. Our list of DEGs was obtained considering a *p*-value < 0.01 and a *q*-value < 0.27, which represent a risk (27%) of a false positive. It has been previously accepted that *q*-value can be higher than 0.05; recent studies using RNA-seq have fixed the *p-*value at <0.05 or <0.01, while *q*-value was cut off at 0.1 [111] or higher than 0.1 [1,22,112,113] to increase DEG detection. One reason why we might optimize DEG detection is that a shortlist of DEGs reduces the significance of the enrichment analysis, limiting the conclusions that can be drawn from the results.

Although the findings should be interpreted with caution, this study has several strengths. It is the first in-depth analysis of jejunal gene expression associated with feed efficiency in slow-growing chicken. Our RT-qPCR results confirmed the differential expression—as shown by RNA-seq—of several of the DEGs examined. Among the genes confirmed, several had a *q*-value between 0.05 and 0.26. Our list of enriched GO-terms and pathways were consistent with previous literature in poultry. Our study represents the first molecular portrait of the jejunal genes influencing feed efficiency in slow-growing chicken.

It will be helpful for optimizing the method of detection of genes associated with feed efficiency. Future studies should perform the experiments described here with an increased sample size and should integrate different critical parameters (gene-specific mean counts and dispersion) revealed by the present study. Another possible area of future research would be to investigate and confirm the candidate genes highlighted in our study. This study opens up prospects for more in-depth investigations.

## 5. Conclusions

This is the first RNA-seq analysis of differential gene expression in the jejunal tissues of high- and low-FCR slow-growing chickens. The DEGs expressed more highly in the low-FCR group are associated mainly with the regulation of immune response activation, and specifically, of macrophages, T-cell, apoptosis, and necroptosis pathways; some are associated with the regulation of glutathione metabolism, the transport and processing of vitamins A, B8, and B12, and the metabolism of lipids, including the beta-oxidation of fatty acids. Others are associated with neuronal development and maturation. However, genes that promote FCR in KR chickens may negatively affect body weight control and thermoregulation, reducing their potential to adapt to changes in feed intake or environmental temperature. It would be interesting to assess the effects of FCR selection on body weight control and thermoregulation. Notwithstanding the relatively limited sample size, these findings provide an important molecular basis for future breeding strategies to improve slow-growing chicken feed efficiency.

## Figures and Tables

**Figure 1 animals-11-02606-f001:**
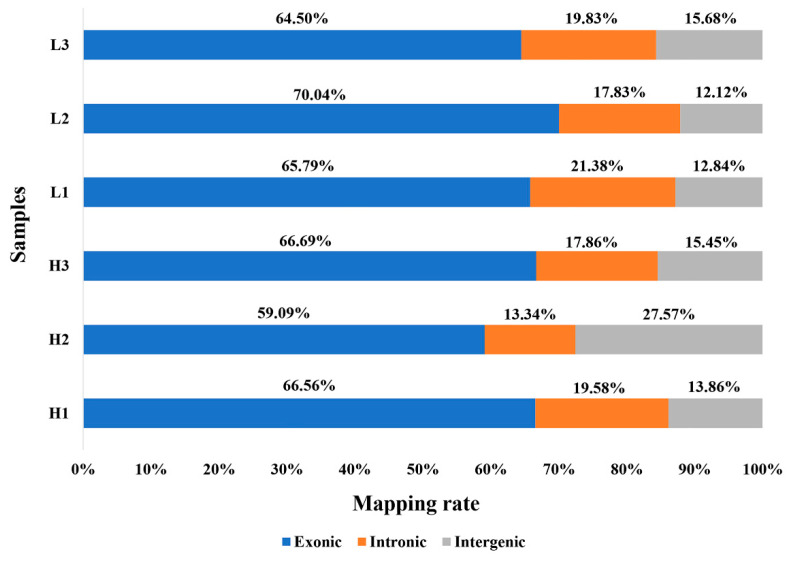
Percentage of reads mapped to exonic, intronic and intergenic regions of the referenced genome.

**Figure 2 animals-11-02606-f002:**
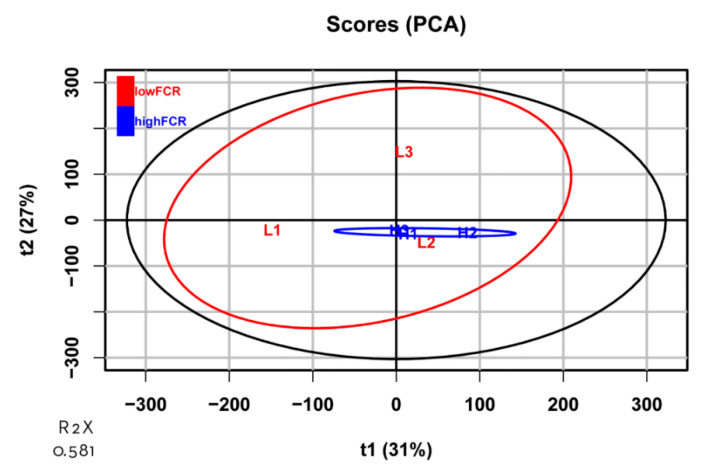
Principal component analysis (PCA) plot of the transcripts identified in high- and low-FCR samples. The ellipses represent 95% confidence intervals for each group. No outliers were observed. The abscissa and ordinate each represent a principal component, and the percentage represents the amount of variation between samples. R2X represents the accumulated variance contribution rate.

**Figure 3 animals-11-02606-f003:**
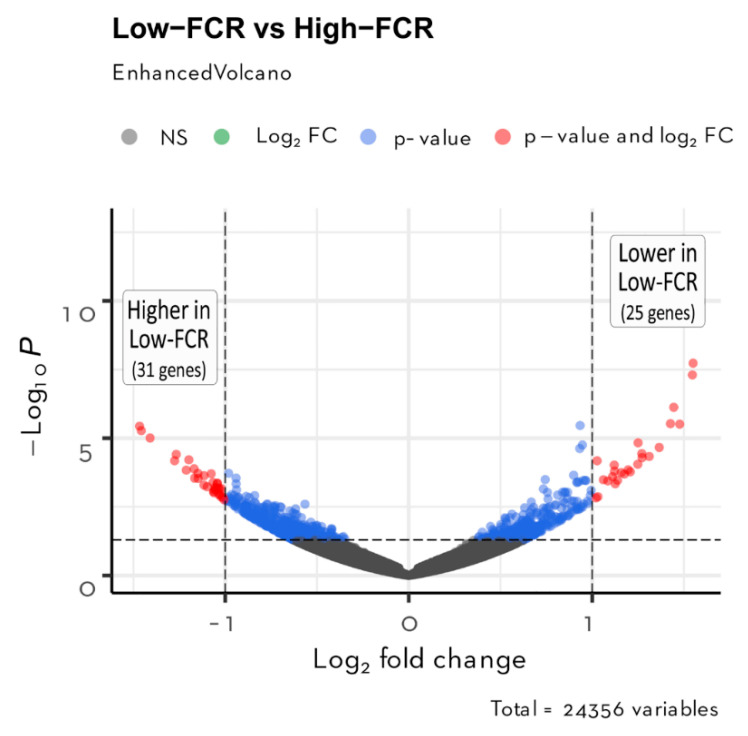
Volcano plot of the 24,356 transcripts quantified in the jejunum from KR chickens with differences in FCR. The horizontal lines indicate the significant thresholds of DEGs at *p*-value < 0.01. The vertical line corresponds to the threshold of |log_2_ FC| > 1. Red dots represent the significant DEGs at *p*-value < 0.01 and |log_2_ FC| > 1**.**

**Figure 4 animals-11-02606-f004:**
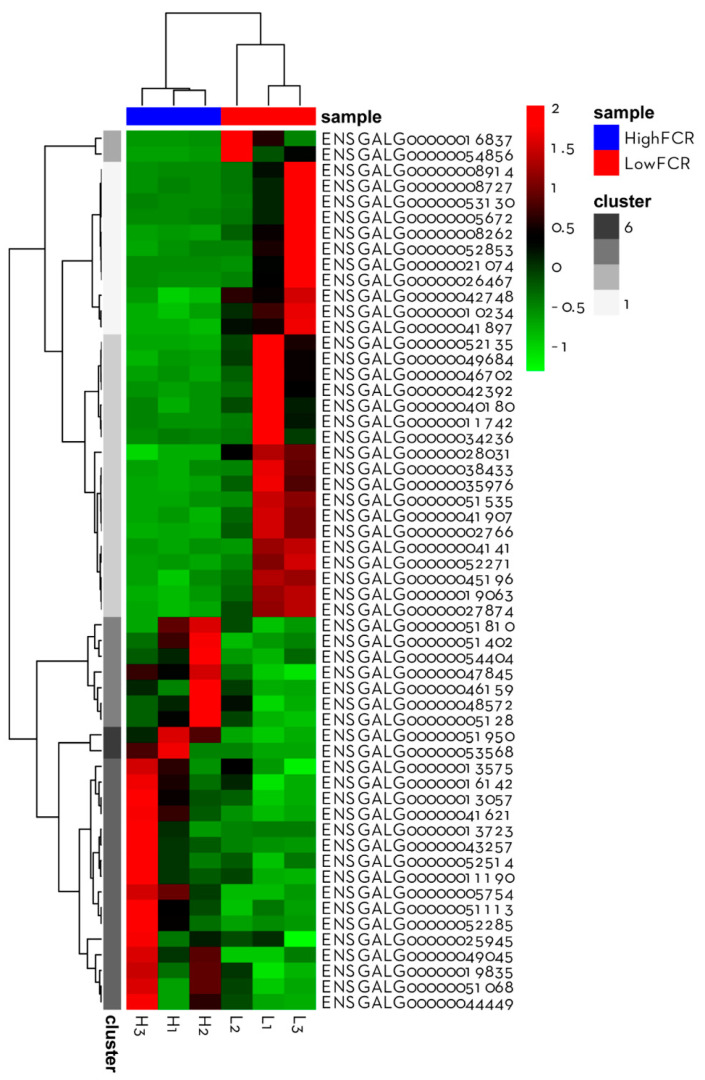
Heatmap featuring the gene expression profile of all genes associated with FCR. Each column represents an individual KR chicken, sorted by group (high- and low-FCR). Each row corresponds to a differentially expressed gene, and red and green colors indicate respectively higher and lower expression relative to average expression across all samples.

**Figure 5 animals-11-02606-f005:**
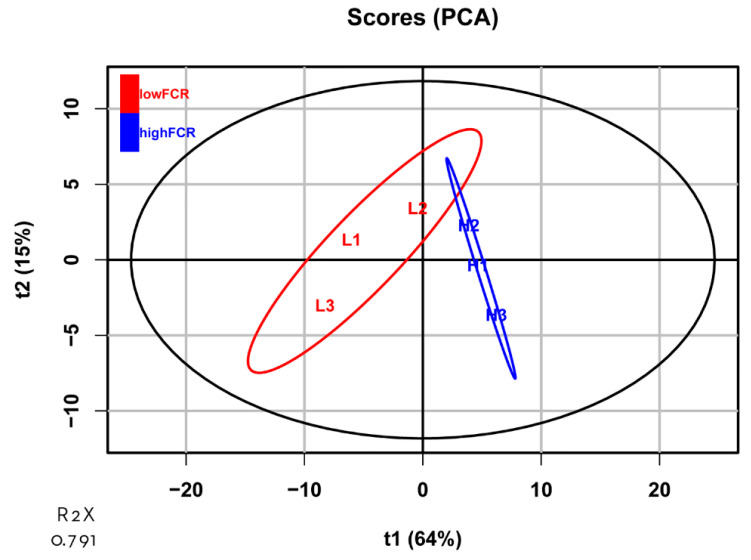
Principal component analysis (PCA) plot of all genes associated with FCR. The ellipses represent 95% confidence intervals for each group (high-FCR, blue; low-FCR, red). The abscissa and ordinate each represent a principal component, and the percentage represents the amount of variation between samples. R2X represents the accumulated variance contribution rate.

**Figure 6 animals-11-02606-f006:**
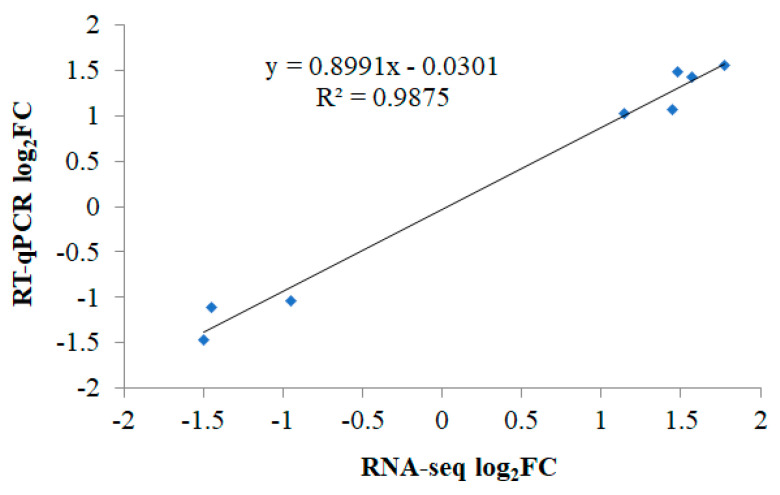
Regression analysis illustrating the correlations between the RNA-seq and RT-qPCR log_2_ FC scores for selected DEGs.

**Figure 7 animals-11-02606-f007:**
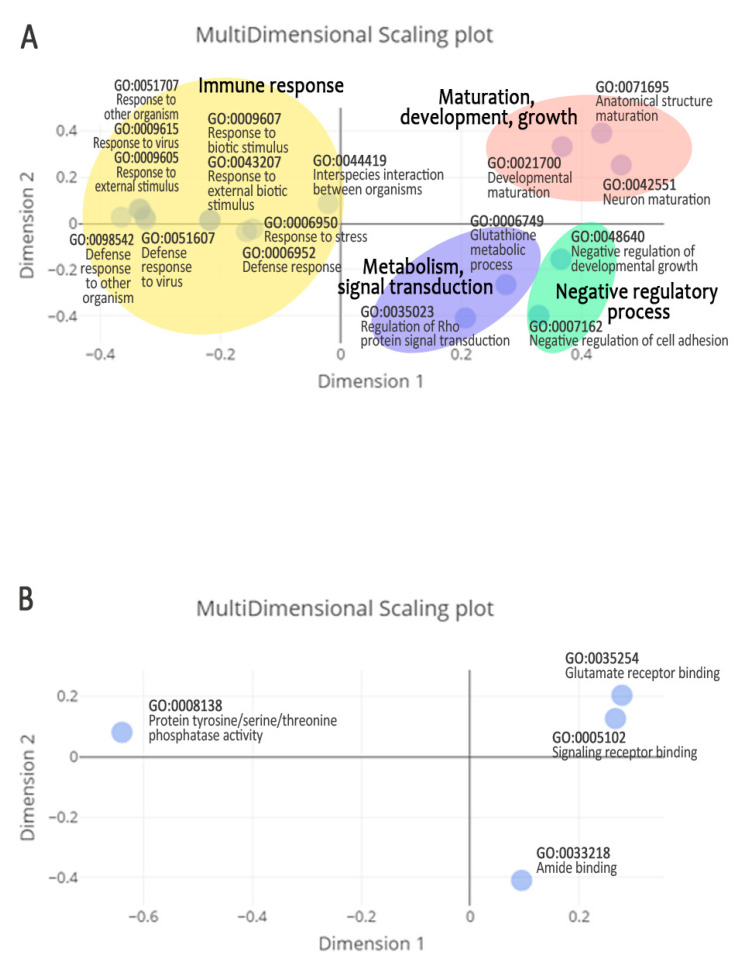
Multi-dimensional scaling (MDS) plots of Ensembl-derived GO terms associated with differences in FCR. Two data features are considered: biological process (**A**) and molecular function (**B**). The MDS plots provide a two-dimensional distance map of the Ensembl-derived GO terms associated with difference in FCR. The units of dimension 1 and dimension 2 are arbitrary. Each point represents a GO term ID with an associated name. Points that cluster nearer to one another in an MDS plot are more similar to one another. Colored areas are drawn to highlight the clusters, and thus the main biological processes associated with difference in FCR.

**Figure 8 animals-11-02606-f008:**
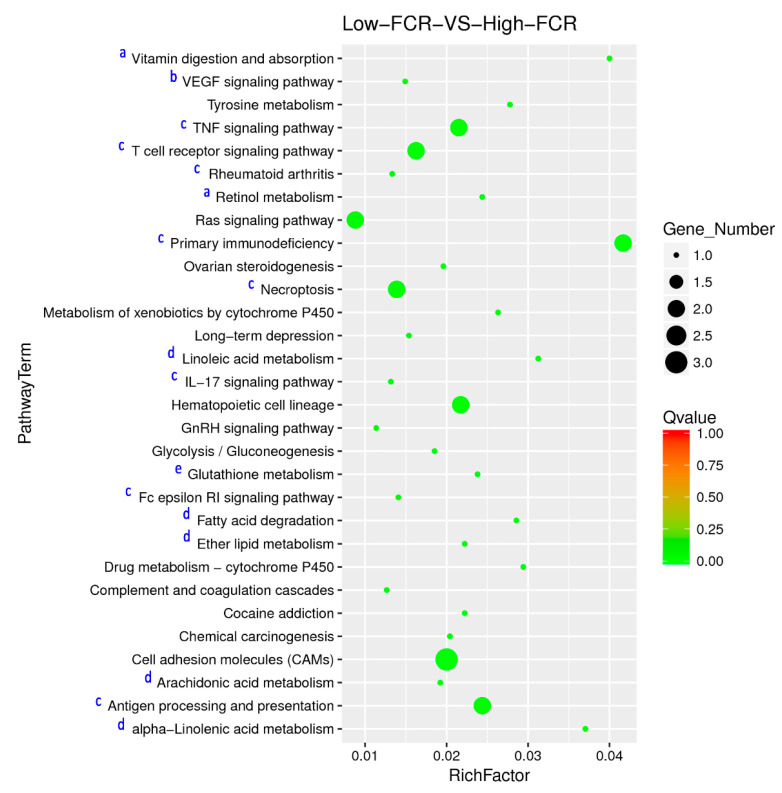
Scatter plot showing the top 30 of enriched KEGG pathways associated with difference in FCR. The size of the dots is positively correlated to the number of DEGs that belong to the pathway. The main pathways including vitamin metabolism (a), neuronal and cardiac maturation, development and growth (b), immune response (c), lipid metabolism (d) and glutathione metabolism (e) are indicated in the Y axis.

**Table 1 animals-11-02606-t001:** Growth performances of KR chickens with low- and high-FCR from 1 to 10 weeks of age.

Trait	Low-FCR (*n* = 3)	High-FCR (*n* = 3)	*p*-Value
FI (g)	3057.72 ± 209.45	3958.28 ± 217.62	0.04
BWG (g)	1580.00 ± 84.61	1204.67 ± 86.15	0.03
FCR	1.93 ± 0.05	3.29 ± 0.06	<0.01

Abbreviations: FI, feed intake; BWG, body weight gain; FCR, feed conversion ratio values represent the means ± SEM.

**Table 2 animals-11-02606-t002:** Summary of data filtration, sequence quality and alignment for jejunal transcriptome of KR chickens with differences in feed conversion ratio.

Group	Low-FCR			High-FCR		
**Sample Name**	**L1**	**L2**	**L3**	**H1**	**H2**	**H3**
Raw reads	46,556,586	42,441,550	41,862,714	43,219,544	45,950,336	44,850,740
Clean data (filtered)	45,538,686	41,586,560	41,004,692	42,344,828	45,144,392	43,888,094
Clean bases	6,713,094,704	6,135,524,782	6,043,751,646	6,241,855,177	6,662,269,608	6,469,832,206
Q20 (%)	96.47	96.79	96.80	96.84	96.94	96.67
Q30 (%)	91.70	92.18	92.25	92.32	92.41	91.98
GC content (%)	48.25	48.95	48.77	48.56	50.10	49.08
Total mapped reads	37,896,269	35,507,285	34,761,891	35,814,976	39,049,374	36,926,584
Multiple mapped reads	2,307,429 (5.06%)	2,033,958 (4.89%)	2,016,644 (4.91%)	2,016,811 (4.76%)	3,541,824 (7.84%)	2,263,424 (5.15%)
Unique mapped reads	35,588,840 (78.15%)	33,473,327 (80.49%)	32,745,247 (79.85%)	33,798,165 (79.81%)	35,507,550 (78.65%)	34,663,160 (78.98%)
Spliced mapped reads	9,275,035	10,151,196	9,217,131	9,763,110	10,550,050	9,865,230
Mapping rate (%)	83.21	85.38	84.77	84.57	86.49	84.13

Abbreviations: FCR, feed conversion ratio. Q20 and Q30 represent the proportion of bases with a Phred quality score greater than 20 and 30, respectively. GC represents the GC content of the clean data.

**Table 3 animals-11-02606-t003:** Genes differentially expressed (*p* < 0.01; |log_2_ FC| > 1) in the jejunum of KR chickens with differences in feed conversion ratio.

**Gene ID ^a^**	**Gene Symbol**	**Chr**	**log_2_ FC (High/Low)**	***p*-Value**	***q-*Value**
** *Genes upregulated in Low-FCR (31 genes)* **					
ENSGALG00000008262	*RASGRF1*	10	−1.01	1.75 × 10^−3^	0.263
ENSGALG00000021074	*MYOC*	8	−1.01	1.42 × 10^−3^	0.236
ENSGALG00000019063	*MMP10*	1	−1.02	6.95 × 10^−4^	0.170
ENSGALG00000046702	*-*	1	−1.02	1.40 × 10^−3^	0.236
ENSGALG00000004141	*LBFABP*	23	−1.03	1.08 × 10^−3^	0.211
ENSGALG00000011742	*ART4*	1	−1.03	7.36 × 10^−4^	0.171
ENSGALG00000042748	*-*	22	−1.03	1.16 × 10^−3^	0.220
ENSGALG00000045196	*LAPTM4B*	2	−1.04	6.45 × 10^−4^	0.161
ENSGALG00000051535	*-*	13	−1.04	4.34 × 10^−4^	0.123
ENSGALG00000026467	*MESP1*	10	−1.04	4.49 × 10^−4^	0.123
ENSGALG00000038433	*SPON2*	4	−1.05	7.21 × 10^−4^	0.170
ENSGALG00000041907	*CDK5R1*	27	−1.06	6.34 × 10^−4^	0.161
ENSGALG00000028031	*C1QL1*	27	−1.06	7.63 × 10^−4^	0.172
ENSGALG00000052853	*-*	KZ626838.1	−1.06	9.61 × 10^−4^	0.191
ENSGALG00000016837	*MYO16*	1	−1.06	8.80 × 10^−4^	0.191
ENSGALG00000052135	*-*	6	−1.07	4.11 × 10^−4^	-
ENSGALG00000049684	*SPAG1*	2	−1.08	1.97 × 10^−4^	0.087
ENSGALG00000053130	*MYOZ3*	13	−1.10	5.83 × 10^−4^	0.151
ENSGALG00000027874	*CHAC1*	5	−1.12	2.32 × 10^−4^	0.094
ENSGALG00000008727	*PLA2G4B*	5	−1.12	5.10 × 10^−4^	0.135
ENSGALG00000034236	*-*	33	−1.15	2.92 × 10^−4^	0.105
ENSGALG00000054856	*LOC100857280*	4	−1.15	1.90 × 10^−4^	0.087
ENSGALG00000052271	*SLC23A3*	7	−1.17	2.85 × 10^−4^	0.105
ENSGALG00000010234	*MMACHC*	8	−1.17	1.30 × 10^−4^	0.077
ENSGALG00000040180	*CEBPD*	2	−1.20	6.13 × 10^−5^	0.046
ENSGALG00000008914	*NRAP*	6	−1.21	1.46 × 10^−4^	0.080
ENSGALG00000035976	*-*	22	−1.27	3.89 × 10^−5^	0.036
ENSGALG00000042392	*-*	13	−1.28	6.62 × 10^−5^	0.046
ENSGALG00000041897	*YJEFN3*	28	−1.41	9.85 × 10^−6^	0.015
ENSGALG00000005672	*SLC6A7*	13	−1.46	5.32 × 10^−6^	0.009
ENSGALG00000002766	*MLKL*	11	−1.47	3.65 × 10^−6^	0.007
** *Genes downregulated in Low-FCR (25 genes)* **					
ENSGALG00000048572	*LOC771880*	4	1.02	1.49 × 10^−3^	0.240
ENSGALG00000025945	*AVD*	Z	1.03	6.67 × 10^−5^	0.046
ENSGALG00000046159	*KNG1*	9	1.03	1.35 × 10^−3^	0.235
ENSGALG00000013575	*IFI6*	2	1.06	3.27 × 10^−4^	0.109
ENSGALG00000051950	*-*	KZ626836.1	1.09	3.59 × 10^−4^	-
ENSGALG00000019835	*TRIM27.2*	16	1.11	2.52 × 10^−4^	0.099
ENSGALG00000052514	*LOC107055361*	28	1.12	9.53 × 10^−5^	0.059
ENSGALG00000013723	*OASL*	12	1.12	1.59 × 10^−4^	0.084
ENSGALG00000051810	*-*	20	1.13	4.57 × 10^−4^	0.123
ENSGALG00000051402	*-*	Z	1.14	3.43 × 10^−4^	0.109
ENSGALG00000005128	*PTPDC1*	12	1.16	1.79 × 10^−4^	0.087
ENSGALG00000005754	*-*	11	1.18	2.03 × 10^−4^	0.087
ENSGALG00000051113	*LOC107053928*	8	1.20	1.42 × 10^−4^	0.080
ENSGALG00000051068	*-*	4	1.21	1.67 × 10^−4^	0.085
ENSGALG00000043257	*LYGL*	1	1.25	8.90 × 10^−5^	0.058
ENSGALG00000044449	*LAG3*	1	1.25	1.47 × 10^−5^	0.020
ENSGALG00000016142	*MX1*	1	1.27	3.62 × 10^−5^	0.035
ENSGALG00000049045	*-*	Z	1.27	5.08 × 10^−5^	0.041
ENSGALG00000054404	*-*	24	1.31	4.61 × 10^−5^	0.039
ENSGALG00000053568	*LOC107049158*	4	1.37	2.17 × 10^−5^	0.025
ENSGALG00000011190	*PLAC8*	4	1.43	2.96 × 10^−6^	0.007
ENSGALG00000047845	*-*	1	1.44	7.49 × 10^−7^	0.003
ENSGALG00000052285	*LOC771880*	4	1.48	3.08 × 10^−6^	0.007
ENSGALG00000013057	*USP41*	1	1.55	4.96 × 10^−8^	3.40 × 10^−4^
ENSGALG00000041621	*LY6E*	2	1.55	1.86 × 10^−8^	2.55 × 10^−4^

Abbreviations: Chr, chromosome; FC, fold change; FCR, feed conversion ratio. ^a^ Identification of genes according to Ensembl database.

**Table 4 animals-11-02606-t004:** Entrez Gene-derived GO terms associated with differences in FCR.

GO ID	Term	Gene Frequency	*p*-Value	Gene Symbol	Category
GO:0003161	Cardiac conduction system development	20% (1/5)	0.008	*MESP1*	BP
GO:0009235	Cobalamin metabolic process	20% (1/5)	0.008	*MMACHC*	BP
GO:0032911	Negative regulation of transforming growth factor beta1 production	20% (1/5)	0.008	*LAPTM4B*	BP
GO:0051901	Positive regulation of mitochondrial depolarization	20% (1/5)	0.008	*MYOC*	BP
GO:1904181	Positive regulation of membrane depolarization	20% (1/5)	0.008	*MYOC*	BP
GO:0019842	Vitamin binding	4% (2/50)	0.006	*AVD*, *MMACHC*	MF
GO:0033218	Amide binding	2.239% (3/134)	0.003	*AVD*, *MMACHC*, *LAPTM4B*	MF

Abbreviations: BP, biological process; MF, molecular function.

## Data Availability

Illumina sequencing raw reads data and processed data have been uploaded to NCBI Gene Expression Omnibus (GEO) database, accession number: GSE178916.

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
