# Peer review of "Jejunal Transcriptomic Profiling for Differences in Feed Conversion Ratio in Slow-Growing Chickens"

_animals, 2021, doi:10.3390/ani11092606_

Round 1

Reviewer 1 Report

The manuscript is well written and the results are 

well described. However, transcriptome analysis 

is based on a total of 6 samples (3 vs 3), too small
number of samples to provide serious scientific

speculations. 

Also, DEG analysis showed that only few genes 

were overrepresented by significant KEGG  

and GO Term. Since the RNA was isolated

from the entire section of jejunum, It is not possible to assign which “part” is associated

 to significant gene expression variation. For instance, the

variation of expression of lipid metabolism

related genes could be only associated to epithelial 

cells. 

This is a pilot study, and a larger population
should be considered to confirm the results of this study. 

The Authors could include the limits of the study 

(small population, entire jejunum, ecc) in both 

Discussion and Conclusions sections. 

Reviewer 2 Report

In this manuscript, Sinpru et al reported jejunal transcriptomic difference between slow-growing chickens of high and low reed conversion ratio (FCR) using RNA-seq followed by functional analysis. The authors claimed that some biological processes, including immune response and metabolism process, potentially affect FCR of slow-growing chickens. The manuscript is well written, and their conclusions are largely consistent with previous studies on feed efficiency in chicken and other domestic animals. However, given that only 56 genes are significantly differentially expressed (|log2FC| > 1 and adjust p-value < 0.05) between the two groups of chickens of divergent FCR, the GO-terms and KEGG pathways-based over-representation analysis of differentially expressed genes is not appropriate. Instead, the gene set enrichment analysis (Subramanian et al. 2005; https://www.pnas.org/content/102/43/15545) which can account for small but consistent changes of gene expression is a better choice for this case. In addition, a few minor issues need addressing.

  • In Introduction, the authors might talk about what have been done on poultry feed efficiency.
  • Line 132, “Hisat2” should be “HISAT2”.
  • Lines 140-141, principal component analysis (PCA) by itself is not a method for clustering analysis, even though it can reveal sample similarities/differences.
  • Line 146, since you differential analysis was conducted on the gene level, “eight transcripts” might be better changed to “eight genes”.
  • Lines 152-158,  the authors mentioned they checked the amplification efficiency of PCR primers and the 2^(-deltadeltCt) method is used for qPCR data analysis. Can the authors disclosed the amplification efficiency? Are the amplification efficiency close to 100%? Otherwise, the 2^(-deltadeltCt) method is not valid.
  • Table 2, the row “Clean bases” needs reformatting.
  • Lines 210-213, the authors stated that “…, whereas low-FCR samples 211 were scattered, reflecting natural biological variation in gene expression in the low-FCR ” Another possibility is that there were some batch effect during RNA-seq assays. Sample L2 might be an outlier of the low FCR group, even though the authors claimed that no outlier exists in line 224.
  • Lines 231-232, “log2FC > 1” should be “|log2FC| >1”.
  • Figure 4, based on this heatmap, the genes of higher expression levels were not reproducible. A gene was determined as differential expressed mainly due to its very high expression level in a single sample. L2 is very different from L1 and L3. The heatmap displays the Z-score not absolute expression levels. So the statement that “… and the color of each box indicates an increase (red) or decrease (green) in gene expression.” Is not accurate. Instead, red and green colors indicate higher of lower expression relative to average expression across all samples. Can the authors try to reanalyze the RNA-seq data using the surrogate variable analysis (SVA; Leek and Storey, 2007; https://journals.plos.org/plosgenetics/article?id=10.1371/journal.pgen.0030161) method to remove hidden variations?
  • Table 4, here GO terms related to immune response are not shown here.
  • Figure 8, the number of differentially expressed genes associated with a specific KEGG pathway is less than 5 for all cases, so the statistical significance is not reliable. Though data is not shown for GO term analysis, I believe the number of differentially expressed genes associated with a specific GO term is also very small given there are only 56 DEGs. So GSEA is a better choice. The author can treat genes associated a GO term ( or a KEGG pathway) as a gene set.
  • Lines 301-302, The statement that “higher body weight 301 can be achieved with reduced feed intake” is not sound.  Do the authors meant that higher body weight can be achieved by improving feed efficiency?
  • In Discussion, can the authors compare their findings in the context of current understanding of genetic mechanisms underlying feed efficiency in chickens? To date, several RNA-seq studies have been conducted to study feed efficiency in poultry, to name a few, https://www.ncbi.nlm.nih.gov/pmc/articles/PMC6110741/, https://journals.plos.org/plosone/article?id=10.1371/journal.pone.0136765, https://bmcgenomics.biomedcentral.com/articles/10.1186/s12864-020-6713-y, and https://www.frontiersin.org/articles/10.3389/fgene.2021.607719/full.

Reviewer 3 Report

General comments

The paper deals with the  different jejunal transcriptomic profile in slow growing chickens related to feed efficiency. Maximice feed efficiency is a hot topic in animal production, and in this sense this manuscript can provide some new information about the underlaying metabolic mechanisms which occur in animals with and inproved feed efficiency. The work is within the scope of the journal and presented some novelty although is limited to slow growing chickens. The manuscript in general is sound and is well written. The experimental design is correct. I have only some doubts and minor suggestions about the manuscript

Specific comments

Introduction

In the introduction section it should be justified why authors have focused in the jejunal transcriptomic profile and not in other sites of the intestine (duodenum, cecum)

L47: this sentence is conflusing : “selecting for higher FCR reduces the average daily weight gain and feed intake”, how can be this assesment posible if FCR is calculated as the ratio of feed intake to body weight gain????

L73-80: this paragraph should be deleted from the introduction section, since belong to the results. Instead, the objetive of the study should be clearly stated, ans I suggest to rewrite lines 72-73 in order to clarify the aim of the study.

Material and Methods

Authors should include the ethical consideration statement for the use of animals for experimental pourposes and the number of the approval protocol.

L85: I suggest to include in tables, maybe as supplemental material, the information of the ingredients and chemical composition of the diets

L89: As intake and average daily gain was measured weekly and FCR calculated also on week basis, it is not clear here if authors ranked animals based on FCR in the 10th week of based on mean FCR of all the experimental weeks.

L89: Why only 3 animals per “group” were chosen?, with a group size of 75 animals it seems feasible to choose a higher number of animals for each of the experimental groups. 3 is the minimun number necessary to perform statistical analyses but results are to some extent limited with a small number of replicates. Authors should mention the limitation of the sample size in the discussion of the results and in the conclusions.

L108: delete “Chaicharoenaudomrung et al, 2020” and only leave the reference number

L129: include a reference for Cutadapt

L142: include a reference for EnhancedVolcano

L143: include a reference for pheatmap

L159: include the package or statistical programme used to perform the regression analysis (reference)

Results

Table 1: include in the title the period to which the data correspond. Are they the mean values for the 10 weeks?. Also check if BWG is actually the BWG or the body weight at the end of the experiment

Table 2: include FCR description in the footnote. The lines of “total mapped reads” and “Mapping rate” give the same information. Delete one

Table 3: Do not use abreviations in the title. “genes differentially expressed (p≤0.05) and with Log2FoldChange > │1│. Use the same terms as in the text (up eta downregulated) instead of higher or lower expressed. Describe FCR in the footnote

L256: It is not clear from here onwards if the functional annotations was done only for upregulated genes or also information abouy downregulaed genes is described. It seems that only information about upregutales genes is available, and in my oppinion it is also of importance the genes that are downregulated.

Discussion

L347: “which would not occur to the same extent in the high-FCR group”

L401: “the” instead of “our”

L410: here there is a mistake since here authors said that AVD and SPON2 both had higher expression in the low-FCR group, but AVD was more highly expressed in the high-FCR group (L401)

Reviewer 4 Report

The authors of "Jejunal Transcriptomic Profiling for Differences in Feed Conversion Ratio in Slow-Growing Chickens" presented the results of an RNA-Sequencing analysis of jejunal tissue between low- and high-feed conversion ratio (FCR) in KR chickens. The methodology applied in the current study scientifically sounds and the study has the potential to contribute to a better understanding of the biological processes associated with feed efficiency in chickens. However, there are serious concerns that must be clarified and fixed (if possible) prior to publication. Check my detailed comments below.

Major comments

The experimental design presented in the study scientifcally sounds. However, it is neecssary to be careful with the interpretation of the results. The authors used 3 samples in each group. The groups were defined based a complex trait (FCR), which can me influenced by several enviromental and genetic components. Consequently, there is a significant chance that the authors dont have enough statistical power to detect true differences in the expression. This is reinforced by the cluster analysis showed in Figure 2. I would strongly recommend the authors to perform a estimation of the detection power of the experiment. There are several tools to perform this kind kind of estimation and it will helps to reinforce the obtaine results.

Additionally, in the material and methods section, the authors informed that the p-values were adjusted using a Benjamini-Hochberg correction. However, in the tables the authors are reporting raw p-values. Are the values showed in the tables really raw p-values? If yes, the results and discussion must be rewritten. The filter to identify DEG is based on p ≤ 0.05 and log2 FC > 1. If raw p-values are used for this selection, the chance of false-positives increase substantially. In addition, the log2 FC > 1 is a very weak signal of biological evidence. Consequently, this reinforce my concern about the use of raw p-values. The authors must present a supplementary material with all the results of the DEG with raw p-values, adjusted p-values, log2FC, test statistics, etc. for all the genes. 

The enrichemnt analysis for GO terms and KEGG pathways is also a concern. It seems that the authros are also reporting raw p-values. The enriched terms and pathways must be defined using adjusted p-values. All the discussion is based on the enriched terms and pathways. Therefore, it is not acceptable that the results of the enrichment analysis is inflated by false-positive results. Additionally, it is interesting to cite that the discussion in several parts is very speculative and most relevant links between the processes and the feed efficiency status should be provided.

Minor comments

Lines 45-48: These sentences are confusing. The two sentences seems to be even controversial.

Lines 129-131: Ok. But which threshold was used to filter the data (Phred 20 or 30)? A complete description of data filtering must be provided.

Lines 146-147: Which transcripts?

Lines 178-180: This sub-section should not be placed at the end of the material and methods section.

Line 266: Different font size in the table title.

Line 337: List the genes.

Lines 363-364: Which kind of genome-wide analysis?

Round 2

Reviewer 1 Report

The manuscript was already well written, but it

has been improved considerably.
In my honest opinion, It is suitable

for publishing.